# Isolation and Characterization of a Bacteriophage with Potential for the Control of Multidrug-Resistant *Salmonella* Strains Encoding Virulence Factors Associated with the Promotion of Precancerous Lesions

**DOI:** 10.3390/v16111711

**Published:** 2024-10-31

**Authors:** Luis Amarillas, Fedra Padilla-Lafarga, Rubén Gerardo León Chan, Jorge Padilla, Yadira Lugo-Melchor, Jesús Enrique López Avendaño, Luis Lightbourn-Rojas, Mitzi Estrada-Acosta

**Affiliations:** 1Instituto de Investigación Lightbourn, Jimenez 33981, Mexico; l.amarillas@institutolightbourn.edu.mx (L.A.); r.leon@institutolightbourn.edu.mx (R.G.L.C.); 2Facultad de Agronomía de la Universidad Autónoma de Sinaloa, Culiacán 80000, Mexico; 3Centro de Investigación y Asistencia en Tecnología y Diseño del Estado de Jalisco, Guadalajara 44270, Mexico

**Keywords:** antibiotic resistance, wastewater epidemiology, oncogenic *Salmonella*, lytic phage

## Abstract

Background: Antimicrobial-resistant bacteria represent a serious threat to public health. Among these bacteria, *Salmonella* is of high priority because of its morbidity levels and its ability to induce different types of cancer. Aim: This study aimed to identify *Salmonella* strains encoding genes linked to the promotion of precancerous lesions and to isolate a bacteriophage to evaluate its preclinical potential against these bacteria. Methodology: An epidemiological approach based on wastewater analysis was employed to isolate *Salmonella* strains and detect genes associated with the induction of precancerous lesions. Antimicrobial susceptibility was assessed by the disk diffusion method. A bacteriophage was isolated via the double agar technique, and its morphological characteristics, stability, host range, replication dynamics, and ability to control *Salmonella* under different conditions were evaluated. The bacteriophage genome was sequenced and analyzed using bioinformatics tools. Results: Thirty-seven *Salmonella* strains were isolated, seventeen of which contained the five genes associated with precancerous lesions’ induction. These strains exhibited resistance to multiple antimicrobials, including fluoroquinolones. A bacteriophage from the *Autographiviridae* family with lytic activity against 21 bacterial strains was isolated. This phage exhibited a 20 min replication cycle, releasing 52 ± 3 virions per infected cell. It demonstrated stability and efficacy in reducing the *Salmonella* concentration in simulated gastrointestinal conditions, and its genome lacked genes that represent a biosafety risk. Conclusion: This bacteriophage shows promising preclinical potential as a biotherapeutic agent against *Salmonella*.

## 1. Introduction

Currently, bacterial infections account for one in every eight deaths worldwide, and this figure continues to rise [1]. Moreover, this problem is aggravated by the emergence of bacteria that are multidrug-resistant to antimicrobials [2]. In the absence of effective preventive measures, projections suggest that this type of bacteria could become the leading cause of death in the coming years [3]. Consequently, experts have emphasized the urgent need for immediate intervention to mitigate this escalating crisis [4,5].

Among the particularly dangerous multidrug-resistant bacteria, considered priority pathogens by the World Health Organization, *Salmonella* stands out for its high morbidity rate [6]. Worldwide, between 200 million and 1 billion infections caused by *Salmonella* are reported annually, leading to approximately 420,000 deaths [7,8,9].

In addition, growing evidence suggests that *Salmonella* is implicated in the development of different types of cancer in the gastrointestinal tract [10,11,12,13]. Although many aspects of *Salmonella*’s role in promoting precancerous lesions and tumorigenesis remain unclear, it is known that the oncogenic potential exhibited by certain *Salmonella* strains is directly attributed to the ability to synthesize certain effector proteins, including AvrA, SopB, CdtB, PltA, and PltB [14,15,16,17]. These proteins contribute chronic inflammation, DNA damage, and manipulation of host signaling pathways [18,19].

This underscores the pressing need for more effective antimicrobial agents to reinforce public health strategies against these types of bacteria. In this context, there is renewed interest in the use of bacteriophages, also called phages, viruses that infect and lyse bacteria, as therapeutic agents [20,21,22]. These viruses represent one of the most promising alternatives for acting against resistant bacteria due to their highly species-specific manner, which enables the selective targeting and destruction of pathogenic bacteria without harming beneficial ones, a crucial advantage in precision medicine [23]. Furthermore, phages are generally recognized as safe, as they do not directly affect human or animal health [24]. Their low production costs also make them an attractive option for biopharmaceutical application [25].

However, not all phages are suitable as therapeutic agents, as some may carry genes encoding pathogenicity factors, which could increase the virulence of the bacteria. Others, meanwhile, may contain proteins with allergenic potential, be unstable during storage, or fail to exert their therapeutic effect at the intended site of action [26,27,28]. Therefore, it is essential to select phages through comprehensive characterization to assess both their therapeutic potential and biosafety.

Accordingly, the main aim of this research was to isolate and characterize a bacteriophage with preclinical potential as an alternative for the treatment for multidrug-resistant *Salmonella* strains, particularly those encoding virulence factors associated with the induction of precancerous lesions.

## 2. Materials and Methods

### 2.1. Isolation of Salmonella Strains and Identification of Genes with Oncogenic Potential

*Salmonella* strains were isolated from 54 urban wastewater samples collected in August–September 2023 from the central zone of the State of Sinaloa, Mexico. Samples were obtained from the sewage system in the municipalities of Culiacan and Navolato. The ultrafiltration method described by Liu et al. (2021) was employed due to its high efficiency in concentrating pathogenic bacteria present in environmental samples, enabling a more effective recovery of bacteria such as *Salmonella* [29]. Subsequently, the bacteria were isolated by inoculating the eluate on Hektoen enteric agar. The presence of the *inv*A gene was confirmed using polymerase chain reaction (PCR), ensuring accurate identification of each bacterial strain [30]. In addition, PCR analysis was also utilized to detect genes associated with the promotion of precancerous lesions, which are encoded by specific *Salmonella* strains. The genes targeted in this study included *avr*A, *sop*B, *cdt*B, *plt*A, and *plt*B. For this purpose, the oligonucleotides and protocols described by lshaheeb et al., 2023, Hawwas et al., 2022, and Mezal et al., 2014 were used for analysis (Table 1) [31,32,33].

### 2.2. Antibiotic Susceptibility Tests

The disk diffusion method was used to evaluate the in vitro susceptibility of the isolated bacterial strains to 12 antibiotics, representing six distinct families and five mechanisms of action. The antibiotics included beta-lactams (ampicillin, amoxicillin/clavulanic acid, cephalothin, imipenem), which inhibit cell wall synthesis; quinolones (ciprofloxacin, nalidixic acid), targeting DNA synthesis; aminoglycosides (gentamicin, amikacin), phenicols (chloramphenicol), and tetracyclines (tetracycline), which inhibit protein synthesis; polymyxins (colistin), which disrupt the cell membrane; and sulfonamides (sulfamethoxazole/trimethoprim), inhibiting folic acid synthesis. This selection covers key biochemical pathways to provide a comprehensive profile of antimicrobial resistance. Antibiotics were chosen based on their clinical relevance in treating *Salmonella* infections and their usage in the local agricultural sector, the predominant activity in the sampling area. For this purpose, 6 mm discs impregnated with standard antibiotic concentrations, as recommended by the International Committee for Laboratory Standards (CLSI), were used. The bacterial inoculums were adjusted to a turbidity equivalent to 0.5 on the McFarland scale, and the plates were incubated at 37 °C for 24 h (h). After incubation, the diameter of the inhibition zones was measured with a calibrated Vernier [34]. *Salmonella* ATCC 14028 was used as the reference strain.

### 2.3. Bacteriophage Isolation and Purification

For bacteriophage isolation, urban wastewater samples were filtered through aluminum oxide nanofibers (NanoCeram-Argonide Corp., Sanford, FL USA), due to their high efficiency in retaining viral particles, using a MasterFlex peristaltic pump (Cole-Palmer, Vernon Hills, Illinois USA) at a constant flow rate of 500 milliliters per minute (mL/min) [35]. The eluate was recovered, and the double-agar-layer technique was used to determine the presence of phages with lytic activity on *Salmonella*. Briefly, 1 mL of the overnight bacterial culture grown in trypticasein soy broth (TSB) was mixed with 200 µL of the eluate and 3 mL of 0.4% TSB-agarose, and the mixture was poured onto Petri dishes containing trypticasein soy agar (TSA). The plates were incubated at 37 °C for 24 h, after which lysis plaques were identified, excised with a Pasteur pipette, and transferred to microcentrifuge tubes containing sterile distilled water. An aliquot of 100 µL was taken, and the double-agar-layer technique was repeated. Plates were again incubated for 24 h at 37 °C [36]. This process was repeated five times to ensure the recovery of a single phage type. The phage was propagated following the protocol described by [37]. Purification of the bacteriophage suspension was achieved via dialysis using the 20,000 MWCO Slide-A-Lyzer system (Thermo Fisher, Waltham, MA, USA), followed by endotoxin removal with the EndoTrap HD system (Lionex, Braunschweig Germany). Subsequently, the suspension was filtered through a polyvinylidene difluoride membrane with 0.22 µm pores.

### 2.4. Transmission Electron Microscopy

For electron microscopy, 30 µL of the purified bacteriophage suspension was placed on a Formvar-coated copper grid with 400-mesh carbon baking and allowed to absorb for 10 min [38]. Subsequently, the grid was placed in a vacuum evaporator (JEE400, JEOL Ltd., Tokyo, Japan), stained with 2% (*w*/*v*) phosphotungstic acid (pH 7.2) for 1 min, and air-dried in a dust-free environment. The samples were examined using a transmission electron microscope (JEM-1011, JEOL Ltd., Tokyo, Japan) operating with an accelerating voltage of 80 kV.

### 2.5. One-Step Replication Curve

The multiplicity of infection (MOI) value of 0.1 was calculated by first determining both the concentration of the bacteriophage and the *Salmonella* strain. The bacteriophage titer was quantified using the double-agar-layer technique in combination with serial dilutions. The concentration of *Salmonella* was assessed by plating on Hektoen enteric agar after the culture reached an optical density of 0.1 at 600 nm.

To evaluate the replication kinetics of the bacteriophage, a single colony of the host strain was resuspended in 50 mL of TSB medium and incubated at 37 °C with shaking at 100 rpm. When the optical density at 600 nm reached 0.1, the phage was added at a multiplicity of infection (MOI) of 0.1. At five-minute intervals, two aliquots were collected from the culture, one treated with chloroform, which lyses the host cells to release intracellular phage particles, and one left untreated. The aliquots were centrifuged at 10,000× *g* for 1 min, and the supernatant was used to determine the bacteriophage concentration through serial decimal dilutions and the double-agar-layer method, as described by [37]. This procedure was performed in triplicate for each time point to ensure the reproducibility and accuracy of the results. The adsorption phase was monitored by determining the rate of bacteriophage attachment to host cells over time, while the eclipse phase was identified as the period between initial infection and the detection of intracellular phage particles. The latency phase was defined as the interval between infection and the release of newly formed phages into the medium. All phases were assessed in triplicate to ensure the reproducibility and accuracy of the results.

### 2.6. Bacteriophage Host Range

The host range of the bacteriophage was evaluated by the agar double-layer method using a purified suspension of the phage at a concentration of 1 × 10^3^ plaque-forming units per milliliter (PFU/mL). The assay included 37 *Salmonella* strains, 8 *Escherichia coli* strains, and 5 probiotic bacterial strains known to be part of human gut microbiota and considered beneficial, including *Bacillus clausii*, *Bacillus coagulans*, *Lactobacillus paracasei*, *Lactobacillus rhamnosus,* and *Limosilactobacillus reuteri*. Double-layer agar plates were prepared for each strain according to the method described in Section 2.3. The plates were inspected for the formation of lysis plaques on the agar, which indicated the lytic activity of the bacteriophage against the tested strains.

### 2.7. Stability of the Bacteriophage

Accelerated stability test. The accelerated stability test of the phage was conducted according to the protocol described by Xu et al. (2023), with modifications outline below [39]. A purified suspension of phage was incubated at 60, 65, and 70 °C, and its concentration was continuously monitored at 1 or 2 h intervals. The data on viral concentration were analyzed using three reaction kinetic models: zero-order (kt = M_0_ − M), first-order (kt = In (M_0_/M)), and second-order (kt = 1/M − M_0_), respectively. The kinetic model that exhibited the lowest coefficient of determination (*r*^2^) was selected for subsequent calculations, indicating the best fit to the experimental data. The thermodynamic temperature (T) and rate constant (*k*) obtained from the selected kinetic model were applied to the Arrhenius equation (ln(*k*) = (*E_a_*/RT) − ln[A], where *Ea* is the activation energy and R is the gas constant. The values of *E_a_* and A were determined experimentally by a nonlinear fit of the viral concentration data. By combining the Arrhenius equation with the selected reaction kinetics model, we calculated the relationship between time (t) and the actual phage concentration (M), which was the model for predicting the phage lifetime. To validate the model’s applicability under realistic conditions, it was tested at 28 °C, with the bacteriophage concentration measured monthly over six months.

Stability in simulated gastrointestinal environment. The simulated gastrointestinal system was composed of three phases: oral phase, gastric phase, and intestinal phase. The composition of each phase is detailed in Table 2. Electrolyte solutions were sterilized at 121 °C for 15 min at one atmospheric pressure. After sterilization, the corresponding enzymes and bile solution, previously filtered with 0.45 μm filters, were added. The phage was then inoculated in the oral phase at a concentration of approximately 1 × 10⁶ colony-forming units per milliliter (CFU/mL), with the final pH adjusted to 7. The mixture was incubated at 37 °C with agitation at 100 rpm for 2 min. Subsequently, the contents of the oral phase were mixed at a 1:1 ratio with the gastric phase, adjusting the final pH to 3, and incubated under the same conditions for 2 h. Finally, the contents of the gastric phase and the intestinal phase were mixed at a 1:1 ratio, adjusting the final pH to 7, and the solution was incubated for 2 h in the described conditions [40]. At each phase of the experiment, the phage concentration was determined through serial decimal dilutions and the agar double-layer method. The results were expressed as means ± standard deviation of the phage concentration, and values of *p* < 0.01 were considered significant.

### 2.8. Bacteriolytic Activity of the Bacteriophage

In culture medium. The *Salmonella* strain, designated Sal-28, that exhibited the highest level of antibiotic resistance and encoded all five evaluated virulence genes was cultured in TSB medium at 37 °C for 24 h. Afterward, 1 mL of this culture was added to four separate flasks, each containing 200 mL of TSB, and incubated at 37 °C in a shaker at 100 rpm. Bacterial growth was monitored by measuring the optical density at 600 nm (OD_600_). When the OD_600_ reached 0.5 (~2 × 10^8^ CFU/mL), varying concentrations of purified bacteriophage suspension were added: 100 µL to the first flask to a final concentration of 1 × 10^7^ PFU/mL, 1 × 10^6^ PFU/mL to the second, and 1 × 10^5^ PFU/mL to the third. The fourth flask served as a control without phage. All cultures were incubated under the same conditions, and optical density measurements were taken hourly [41].

In the simulated gastrointestinal system. The bacteriolytic capacity of the bacteriophage was evaluated using the in vitro system described in Section 2.7, with modifications to incorporate a food model similar to that proposed by Akritidou et al. (2023), designed to promote bacterial survival [42]. To simulate a microbiome, the five bacterial species described in Section 2.6 were used. All strains were mixed in equal proportions and used immediately [43]. In the simulated intestinal fluid, each of the following treatments were inoculated: (1) beneficial bacteria consortium (1 × 10^6^ CFU/mL) + 50 µL of sterile water, (2) *Salmonella* (1 × 10^6^ CFU/mL) + 50 µL of sterile water, (3) beneficial bacteria consortium (1 × 10^6^ CFU/mL) + 50 µL of purified phage suspension (1 × 10^4^ PFU/mL), (4) *Salmonella* (1 × 10^6^ CFU/mL) + 50 µL of the purified phage suspension (1 × 10^4^ CFU/mL), (5) *Salmonella* (1 × 10^6^ CFU/mL) + ciprofloxacin 0.5 mg/mL, and (6) beneficial bacteria consortium (1 × 10^6^ CFU/mL) + ciprofloxacin 0.5 mg/mL. The bacterial concentration of each treatment was quantified using a plate count method based on spreading the sample on Petri plates. Results were presented as the mean concentration ± standard deviation, with *p*-values < 0.01 considered statistically significant.

### 2.9. Genomic Sequencing and Bioinformatic Analysis

Bacteriophage genetic material was extracted according to the proteinase K/SDS protocol [44]. Genomic libraries were generated using the MGIEasy DNA library Prep Universal System, and nucleotide sequencing was conducted on the MGISEQ-2000 platform utilizing the DNBSEQTM (nanoballs DNA) system. Assembly and bioinformatics analysis were performed according to the guidelines proposed by Philipson et al. (2018) and Turner et al. (2021) [45,46]. Random sampling of reads (50,000 to 100,000) was performed using the Seqtk Toolkit tool until a coverage depth of ~100× was achieved. After analyzing the quality of reads with FastQC, low-quality adapters and sequences (Phred index < 30) were removed using Trimmomatic. De novo assembly of reads was performed with SPAdes, employing k-mers of 21, 33, and 55. Open reading frames (ORFs) were identified using Glimmer, Genemark, Genemark.hmm, Genemark S, Prodigal, RAST, and MetaGene. Promoters were identified with PhagePromoter and PHIRE, while FindTerm and RNAold were used to identify Rho-independent terminators. The tRNAs were determined by ARAGORN and tRNAscan-SE. The ability of the phage to establish lysogenic cycles was evaluated using PhageAI and PHACTS. The presence of genes associated with antibiotic resistance was analyzed in the CARD platform (https://card.mcmaster.ca accessed on 17 September 2024) and AMRFinderPlus, while virulence factors were identified through VFDB (www.mgc.ac.cn/VFs/search_VFs.htm accessed on 17 September 2024) and VirulentPred (https://bioinfo.icgeb.res.in/virulent/submit.html accessed on 17 September 2024).

### 2.10. Statistical Analysis

All statistical analyses were conducted using Minitab 19. Normality assumptions were verified with the Shapiro–Wilk test, and homogeneity of variance was assessed with Levene’s test. Differences in bacteriophage stability under simulated gastrointestinal conditions and bacteriolytic activity were analyzed by repeated-measures ANOVA and two-factor ANOVA, respectively, both followed by Tukey’s post hoc test. Significance was set at *p* ≤ 0.01. All assays were performed in triplicate.

## 3. Results

### 3.1. Isolation of Salmonella and Identification of Genes Associated with Cancer Induction

A total of 37 *Salmonella* strains were isolated from the wastewater samples, with an average isolation rate of 43%. The high prevalence of this bacterium suggests a significant occurrence of *Salmonella* in urban wastewater, underscoring the importance of monitoring its presence due to its potential impact on public health risks. These findings are congruent with previous studies that have documented the presence of *Salmonella* in urban wastewater systems as an indicator of fecal contamination and insufficient water treatment efforts [47,48]. Notably, 17 of the 37 isolates harbored five virulence genes associated with potential cancer induction, specifically *avr*A, *sop*B, *cdt*B, *plt*A, and *plt*B (Table 3).

According to numerous studies, *Salmonella* strains that encode these genes produce effector proteins implicated in chronic inflammation and the induction of precancerous lesions. Although several aspects remain unclear, it is known that the AvrA protein modulates the β-catenin signaling pathway, potentially influencing the regulation, differentiation, and proliferation of intestinal epithelial cells [49]. In contrast, SopB promotes intracellular invasion of *Salmonella* and inhibits apoptosis [16,50]. Additionally, the cytolethal distending toxin, composed of CdtB, PltA, and PltB proteins, induces DNA damage, which results in genomic instability and a proinflammatory environment associated with precancerous lesions [51]. Consequently, future research will aim to evaluate the pathogenic potential of these strains in cancer promotion.

The detection of *Salmonella* strains harboring genes associated with cancer induction in wastewater highlights the urgent need for stricter water treatment policies to mitigate the spread of these pathogens and their long-term public health implications.

### 3.2. Antibiotic Sensitivity Tests

The antimicrobial sensitivity of the 37 *Salmonella* strains was evaluated against 12 antibiotics commonly used for treating bacterial infections. The results revealed that four (sal01, sal08, sal27, and sal28) of the strains exhibited resistance or intermediate sensitivity to at least three different classes of antibiotics, leading us to categorize them as multidrug-resistant (Table 3). A significantly higher incidence of resistance was observed for colistin (11 strains), followed by nalidixic acid (9 strains), tetracycline (7 strains), and ampicillin (6 strains). The resistance to ciprofloxacin and colistin, two critical antibiotics used the treating severe *Salmonella* infections, and other bacterial infections, is particularly alarming. These findings are consistent with recent reports from other Latin American regions, which also indicate a rising prevalence of antibiotic-resistant bacterial strains [52,53]. The identification of multidrug-resistant *Salmonella* strains highlights the need for new strategies to combat antimicrobial resistance. In this context, research into therapeutic alternatives, such as the use of bacteriophages, may offer a promising approach for addressing this growing public health challenge [22].

### 3.3. Isolation of the Bacteriophage

A total of seven bacteriophages were isolated from wastewater, and the one producing the largest, clearest, and most well-defined lysis plaques was selected for further study. This bacteriophage was designated as Phylax-28 (from the Greek “guard” or “protector), which is capable of producing clear and well-defined lysis plaques, with a diameter of approximately 1.2 cm (Figure 1A). The plaque size generated by Phylax-28 is notably larger than typical *Salmonella* phages, which are generally reported to form plaques ranging between 0.5 and 3.5 mm in diameter [54,55,56,57].

In addition, Phylax-28 induces the formation of an inhibition halo, a semitransparent zone surrounding the lysis plaques. This phenomenon is consistent with findings by Jurczak-Kurek and coworkers, who noted that phages that produce clear plaques are often associated with a strong lytic activity against bacteria [58]. The appearance of the halo is attributed to the activity of some enzymes, encoded by a certain phage, that degrade the cell wall, thereby enhancing the phage’s potential for bacterial elimination [58].

### 3.4. Bacteriophage Morphology

Analysis of transmission electron micrographs of 20 virions of Phylax-28 revealed that it has an icosahedral, isometric capsid of 28 ± 0.2 nm in diameter, along with a thin, short, non-contractile, and rigid tail of 8.2 ± 0.1 nm in length (Figure 1B). Based on these morphological features and the criteria established by the International Committee on Viral Taxonomy, the bacteriophage Phylax-28 was classified as a new member of the family *Autographiviridae*.

Usually, phages with short tails or absent tails, such as Phylax-28, are typically associated with increased resistance to harsh environmental conditions [59,60]. These morphological traits suggest that Phylax-28 may maintain stability under aggressive biophysicochemical conditions, such as those prevailing in the gastrointestinal system. However, further experimental data are required to confirm this hypothesis. Subsequent sections will detail the experimental results of phage stability under these conditions.

### 3.5. Bacteriophage Replication Curve

Within 5 min of exposure, approximately 80% of Phylax-28 virions were adsorbed on the bacterial surface (Figure 2). Phylax-28 presents a latency period of 15 min, with bacterial lysis occurring at 20 min post-infection, releasing 52 ± 3 virions per bacterial cell.

This burst size observed in Phylax-28 is relatively large compared to other *Salmonella*-phages, which typically produce between 34 and 37 virions per cell [61,62]. Some *Salmonella*-infecting phages, however, have been reported to produce larger burst sizes but with longer latency periods [63,64]. The short latency period of Phylax-28 suggests a competitive advantage over other phages, as they can produce enough virions to lyse the bacterium in a short period of time, making it a promising candidate for *Salmonella* control.

### 3.6. Host Range

Phylax-28 showed the ability to lyse 21 of the 37 *Salmonella* strains and 3 strains of *E. coli*. Bacteriophages capable of lysing two or more bacterial species or diverse strains within a bacterial species, like Phylax-28, are considered to have a broad host range [65]. These bacteriophages are more likely to be chosen as therapeutic agents [66]. However, the therapeutic efficacy of a phage also depends on its specificity. It is essential that phages selectively target pathogenic bacteria without harming bacteria that perform beneficial functions [67]. Notably, Phylax-28 did not exhibit lytic activity against probiotic strains such as *Bacillus clausii*, *Bacillus coagulans*, *Lactobacillus paracasei*, *Lactobacillus rhamnosus,* and *Limosilactobacillus reuteri*, an attribute that could prove advantageous for controlling the population of pathogenic bacteria during infection without affecting beneficial bacteria.

### 3.7. Stability of the Bacteriophage

#### 3.7.1. Storage Stability

An essential criterion for selecting bacteriophages as therapeutic agents is their stability during storage and under the biophysicochemical conditions prevailing at the site of action [39]. For this reason, the storage stability of the Phylax-28 phage was assessed through an accelerated stability assay and validated by monitoring the decrease in phage concentration at 28 °C over six months.

The experimental data on the accelerated stability revealed that the reduction in Phylax-28 concentration followed first-order kinetics, with a correlation coefficient (*r*) of 0.991 at 70 °C (Figure 3).

The coefficient of determination (*r*^2^) for the first-order kinetics was 2.80, higher than the values obtained for the zero-order (2.40) and second-order (2.63) reactions, confirming that the first-order model best described the changes in Phylax-28 concentration. The degradation constant (*k*), calculated from the experimental data obtained for the stability of Phylax-28 at a temperature of 28 °C over six months was estimated to be 0.045 days^−1^, while the activation energy (*Ea*) was 158.6 kJ/mol.

The predictive model suggests that the phage suspension will retain its stable concentration for approximately 92 days at 28 °C before undergoing a 1-logarithm reduction. This prediction is based on the application of the first-order kinetic model together with the Arrhenius equation.

Notably, the experimental results supported the accuracy of this model: after three months of storage at 28 °C, only a 1-logarithm reduction in phage concentration was observed, consistent with model predictions at 94% accuracy. According to Xu et al. (2023) [39], this level of accuracy is considered high. However, the full validation of the model remains ongoing.

#### 3.7.2. Stability in Simulated Gastrointestinal System

Phage Phylax-28 exhibited remarkable stability under simulated gastrointestinal conditions. In the oral phase, no significant reduction in viral concentration was observed, and at the gastric level, the decrease was limited to 1.1 logarithms. Furthermore, the concentration remained stable at the intestinal level in relation to the gastric phase (Figure 4). These finding stand in contrast to those report for coliphage Ace, which experienced a 4-logarithm reduction compared to the initial dose under similar conditions [68]. Additionally, when studying five *Salmonella* phages, Dlamini et al. (2023) documented reductions ranging from 4.86 to 5.55 logarithms, concluding that these phages would require encapsulation in CaCO_3_ for therapeutic viability [69]. Similarly, bacteriophage ZCEC5 showed comparable reductions, with the authors recommending microencapsulation in chitosan–alginate for stability [70]. In comparison, Phylax-28 demonstrated superior stability under gastrointestinal conditions, suggesting its potential as a more robust therapeutic agent.

### 3.8. Capacity of the Bacteriophage to Control Salmonella

#### 3.8.1. In Culture Medium

Under these conditions, Phylax-28 exhibited bacteriolytic activity against *Salmonella*, achieving a significant reduction in bacterial concentration within 60 min of inoculation compared to normal bacterial growth (*p* < 0.01). This effect was observed even at low multiplicities of infection (MOI) of 0.01 and 0.001 (Figure 5). These findings are noteworthy, as many phages demonstrate rapid bacteriolytic activity only at higher MOIs. For instance, Ref. [71] reported a significant reduction in bacterial load only when using MOIs of 10 and 100. Phylax-28’s bacteriolytic activity at low MOIs presents a significant advantage for therapeutic applications, as it suggests that lower phage doses may effectively control *Salmonella* populations. This reduces the potential risk of side effects commonly associated with higher phage doses, offering a safer and more efficient approach to phage therapy.

#### 3.8.2. In Simulated Gastrointestinal System

*Salmonella* tends to colonize and invade the human intestine [72], making it crucial to evaluate the bacteriophage’s ability to control this pathogen under such conditions. In this regard, the results indicated that Phylax-28 significantly reduced the *Salmonella* concentration in a simulated intestinal environment (*p* ≤ 0.01). Moreover, no significant changes were observed in the population of beneficial bacteria due to phage activity (Figure 6).

In contrast, the treatment with ciprofloxacin did not reduce the *Salmonella* concentration compared to the control, and there was a marked reduction in the population of beneficial bacteria caused by the antibiotic. Meanwhile, in the beneficial bacteria, there was a marked reduction in the concentration due to the action of the antibiotic. These findings are highly relevant, as they suggest that Phylax-28 could potentially control *Salmonella* without adversely affecting the microbiome, positioning it as a promising tool in precision medicine. However, further studies are necessary to validate these results.

### 3.9. Bacteriophage Genomics Analysis

The genome of the bacteriophage Phylax-28 consists of linear double-stranded DNA, with a molecular size of 40,989 bp and GC content of 57.81%. It encodes 50 genes, none of which are associated with tRNA. Gene functions were annotated using the NCBI BLAST database (Figure 7).

Gene functions were identified with an e-value of <10^−5^, sequence coverage exceeding 50%, and identity greater than 85%. The genome is organized into distinct functional modules: a morphogenesis module, containing ten genes responsible for the synthesis of virion structural proteins; a packaging module, comprising four genes involved in DNA encapsulation within the capsid; a DNA processing module, with 13 genes regulating the replication and transcription of genetic material; and a lysis module, consisting of three genes responsible for cell wall degradation and bacterial lysis. The remaining genes encode hypothetical proteins. Critically, for biosafety considerations, no genes related to virulence factors, antimicrobial resistance, or allergenic responses were identified. Moreover, Phylax-28 was classified as strictly lytic, a desirable trait for bacteriophages intended for therapeutic applications [73]. The complete genome sequence of phage Phylax-28 was deposited in GenBank/EMBL/DDBJ under the accession number PQ306468.

## 4. Conclusions

Phylax-28 demonstrates substantial potential as a therapeutic agent against multidrug-resistant *Salmonella* strains that encode virulence factors associated with the development of precancerous lesions. In our study, we isolated thirty-seven *Salmonella* strains, of which seventeen were found to encode genes linked to the promotion of these lesions and displayed resistance to multiple antimicrobial agents, including fluoroquinolones. The isolated bacteriophage, belonging to the *Autographiviridae* family, exhibited broad lytic activity against *Salmonella* strains, characterized by a rapid replication cycle and effective reduction in this pathogen’s concentration under simulated gastrointestinal conditions. While further research is necessary to confirm the safety and efficacy of Phylax-28 in clinical settings, preliminary data indicate that it could become a viable biopharmaceutical option for combating antimicrobial-resistant pathogens in the near future. Nevertheless, the reliance on a single bacteriophage poses an increased risk of developing phage-resistant bacterial strains. To mitigate this concern, our research group plans to continue the isolation and characterization of additional bacteriophages, with the goal of developing a multi-phage formulation to enhance the efficacy of *Salmonella* control.

## Figures and Tables

**Figure 1 viruses-16-01711-f001:**
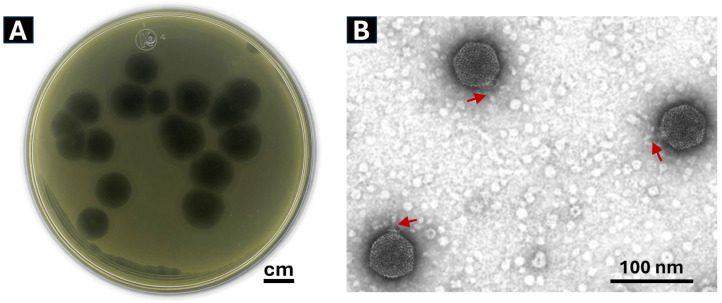
(**A**) Morphology of lysis plaques and (**B**) structural features of Phylax-28 bacteriophage virions. The clarity and size of the plaques reflect the lytic activity and efficacy of the bacteriophage against *Salmonella*. The virions exhibit an icosahedral structure, characterized by a capsid composed of 20 equidistant triangular faces. The phage tails are subtly perceptible and are indicated by the red arrows.

**Figure 2 viruses-16-01711-f002:**
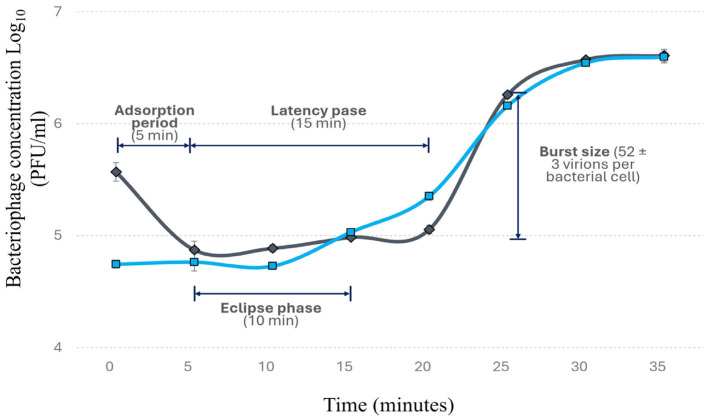
Replication curve of bacteriophage Phylax-28 using *Salmonella* as the host bacterium. The replication dynamics are depicted under two conditions: with chloroform (blue line), which indicates the formation of intracellular virions, and without chloroform (black line), which shows the release of virions into the extracellular medium.

**Figure 3 viruses-16-01711-f003:**
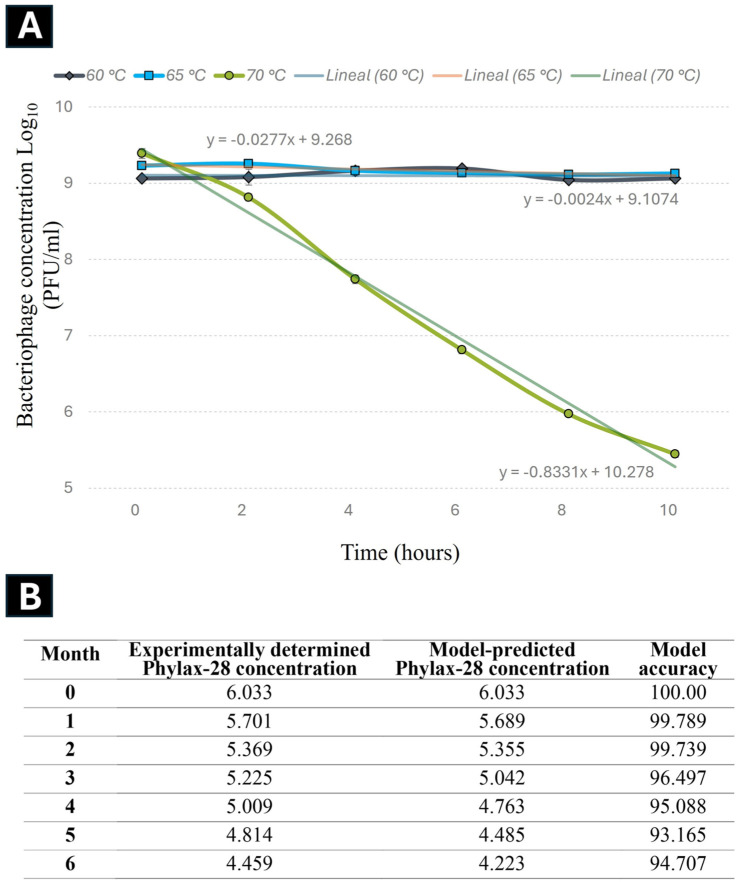
Kinetics of Phylax-28 bacteriophage concentration reduction at various temperatures. (**A**) No significant reduction was observed at 60 °C (black line) and 65 °C (blue line). However, at 70 °C (green line), the reduction follows first-order kinetics with an *r*^2^ value of 0.991. (**B**) Experimental results on the stability of Phylax-28 at 28 °C, measured monthly over six months, were compared with the model’s predicted stability. The phage concentration is expressed as the log10 of PFU/mL. The model’s accuracy was expressed as a percentage.

**Figure 4 viruses-16-01711-f004:**
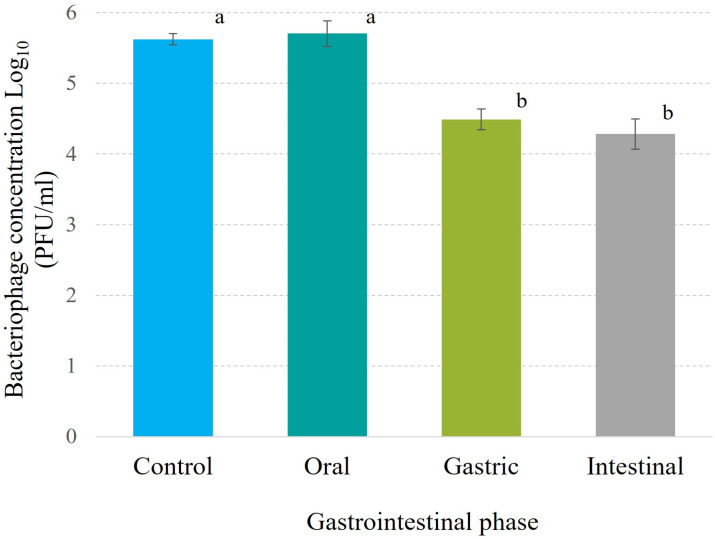
Stability of Phylax-28 bacteriophage under simulated gastrointestinal conditions. In the oral cavity, no significant reduction in phage concentration was observed, indicating high stability in this phase. A slight loss of viability was noted in the gastric environment, but the phage retained functionality. In the intestinal phase, the phage concentration remained stable with no significant changes. The different letters appearing above the bars in the graph indicate that there are statistically significant differences among the treatments. Error bars represent the standard deviation of the measurements, reflecting variability across replicate experiments.

**Figure 5 viruses-16-01711-f005:**
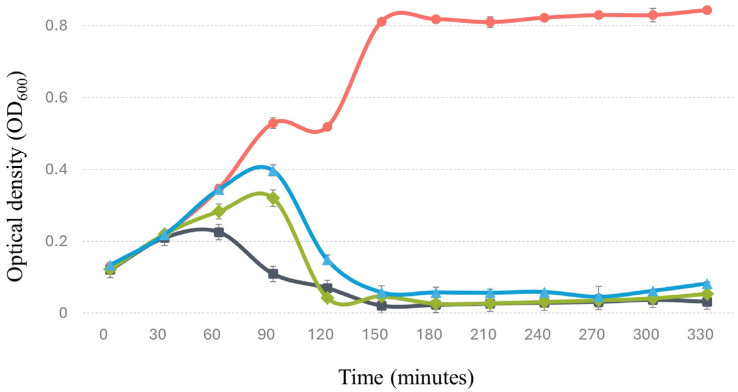
The lytic activity of bacteriophage Phylax-28 against *Salmonella* in trypticase soy broth (TSB). The red line indicates the normal growth of *Salmonella* without bacteriophage treatment, serving as the control. The black, green, and blue lines represent treatments with Phylax-28 at multiplicities of infection (MOI) of 0.1, 0.01, and 0.001, respectively. A significant reduction in bacterial growth is observed across all MOI levels, with Phylax-28 showing pronounced efficacy in reducing *Salmonella* concentrations, even at the lowest MOI.

**Figure 6 viruses-16-01711-f006:**
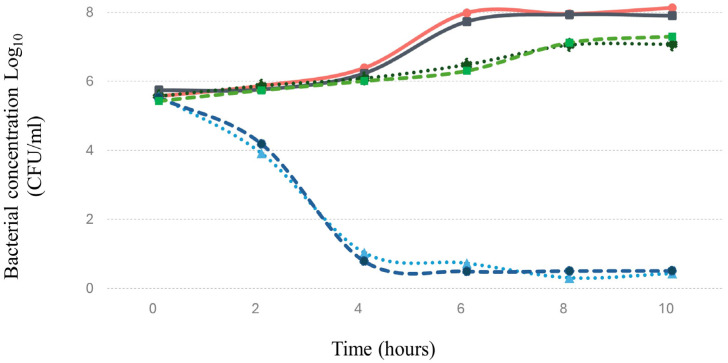
Effect of bacteriophage and ciprofloxacin on the concentration of *Salmonella* and probiotic bacteria in a simulated intestinal environment. The graph displays *Salmonella* growth (red line) in the absence of ciprofloxacin, *Salmonella* in the presence of ciprofloxacin (black line), probiotic bacteria without ciprofloxacin (dashed green line), probiotic bacteria with ciprofloxacin (dashed dark blue line), *Salmonella* in the presence of bacteriophage (dotted light blue line), and probiotic bacteria with bacteriophage (dotted dark green line). The probiotic bacteria used includes five strains: *Bacillus clausii*, *Bacillus coagulans*, *Lactobacillus paracasei*, *Lactobacillus rhamnosus*, and *Limosilactobacillus reuteri*, mixed in equal proportions. The magnitude of error bars, represented as standard deviation, are not discernible on the graph.

**Figure 7 viruses-16-01711-f007:**
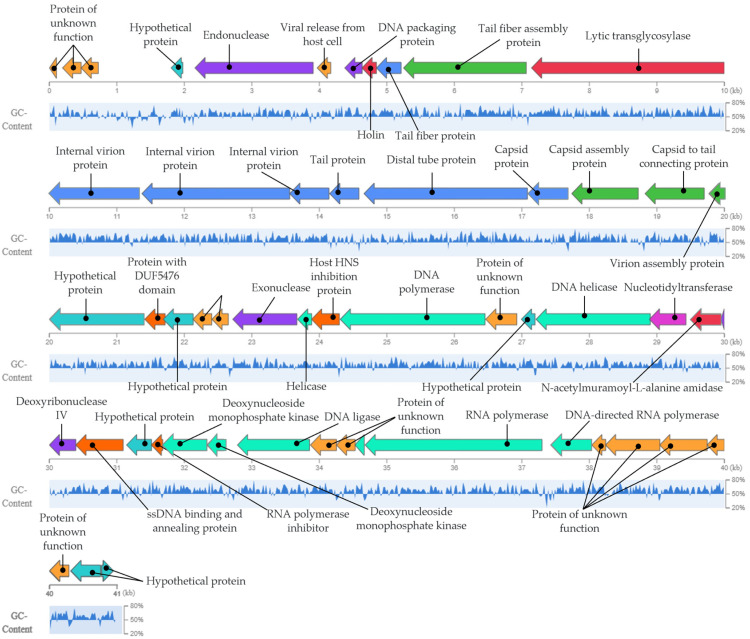
Graphical representation of the genome of bacteriophage Phylax-28. Arrows indicate the genes encoded within the genome, with the function of each gene illustrated by a color. No genes encoding virulence factors, mechanisms for lysogenic cycle establishment, or allergenicity-related genes were identified.

**Table 1 viruses-16-01711-t001:** List of oligonucleotide sequences used for the amplification of genes in *Salmonella* strains encoding effector proteins linked to the promotion of precancerous lesions and tumorigenesis.

Oligonucleotide	Sequence	Gene	Fragment Size (bp)
AvrA-F	CCTGTATTGTTGAGCGTCTGG	*avr*A	422
AvrA-R	AGAAGAGCTTCGTTGAATGTCC		
SopB-F	TCAGAAGRCGTCTAACCACTC	*sop*B	517
SopB-R	TACCGTCCTCATGCACACTC		
CdtB-F	GAAACAAGTCAGGCATTGCC	*cdt*B	819
CdtB-R	GAATGGCTCATAAACACGCC		
PltA-F	GTGGGACTATCATCGTGCAG	*plt*A	729
PltA-R	AGGGTGATCAACGTAACCAC		
PltB-F	GCCGGAAGTACCTGTGTTAT	*plt*B	414
PltB-R	AGTAGTGAAAACCCATCGCG		

**Table 2 viruses-16-01711-t002:** Composition of fluids in each phase of the simulated gastrointestinal tract.

Reagent	Oral Phase (mmol/L)	Gastric Phase (mmol/L)	Intestinal Phase (mmol/L)
KCl	15.1	6.9	6.8
KH_2_PO_4_	3.7	0.9	0.8
NaHCO_3_	13.6	25	85
NaCl	-	47.2	38.4
MgCl_2_(H_2_O)_6_	0.15	0.12	0.33
(NH_4_)_2_CO_3_	0.06	0.5	-
CaCl_2_(H_2_O)_2_	1.5	0.15	0.6
Enzymes			
α-amylase	150 U/mL	-	-
Pepsin	-	4000 U/mL	-
Lipase	-	120 U/mL	-
Pancreatin	-	-	200 U/mL (based on trypsin activity)
Bile salts	-	-	10 mM

**Table 3 viruses-16-01711-t003:** Antimicrobial resistance profiles and virulence factors of *Salmonella* strains isolated from urban wastewater. IMP, imipenem; SXT, trimethoprim–sulfamethoxazole; GEN, gentamicin; AMP, ampicillin; CIP, ciprofloxacin; NAL, nalidixic acid; CHL, chloramphenicol; TET, tetracycline; COL, colistin; AMC, amoxicillin–clavulanate; CFP, cefoperazone; AMK, amikacin. The symbol ‘+’ indicates that the strain carries the virulence gene listed in the table header, while the symbol ‘-’ signifies that the gene was not detected in the strain.

	Antimicrobials and Average Inhibition Diameter (mm)												Virulence Genes Associated with the Potential to Induce Precancerous Lesions				
Bacterial Strain	IMP	SXT	GEN	AMP	CIP	NAL	CHL	TET	COL	AMC	CFP	AMK	*avrA*	*sopB*	*cdtB*	*pltA*	*pltB*
*Salmonella* sal01	26	21	16	15	16	14	24	19	8	21	24	17	-	+	+	-	+
*Salmonella* sal02	26	24	18	19	25	21	25	22	9	10	24	18	+	+	+	+	+
*Salmonella* sal03	22	18	19	16	22	21	17	19	8	17	22	19	+	-	+	+	+
*Salmonella* sal04	25	24	18	14	22	17	22	18	7	18	21	18	+	-	+	-	+
*Salmonella* sal05	29	26	21	22	31	21	23	20	8	22	24	19	+	+	+	+	+
*Salmonella* sal06	26	22	18	21	30	21	23	20	9	22	17	17	+	-	-	-	+
*Salmonella* sal07	27	25	18	21	31	21	31	19	20	17	8	16	+	+	+	+	+
*Salmonella* sal08	15	18	17	15	14	12	21	27	8	22	19	19	+	-	+	+	+
*Salmonella* sal09	29	25	23	31	22	24	29	9	24	19	25	19	+	+	+	+	+
*Salmonella* sal10	26	24	18	23	30	22	26	22	9	22	24	19	+	+	+	-	-
*Salmonella* sal11	26	25	16	21	29	20	24	19	9	22	26	19	+	+	+	+	+
*Salmonella* sal12	26	24	18	21	25	21	25	22	9	23	24	18	+	+	+	+	+
*Salmonella* sal13	22	22	19	21	30	21	24	19	9	21	22	19	+	-	+	-	+
*Salmonella* sal14	25	23	18	22	32	22	22	18	9	22	28	18	+	-	-	+	+
*Salmonella* sal15	29	20	21	22	31	21	23	20	8	22	24	19	+	+	+	+	+
*Salmonella* sal16	26	24	18	21	30	21	23	20	9	22	23	17	+	+	+	+	-
*Salmonella* sal17	21	19	12	21	22	21	31	21	25	18	8	22	+	-	+	+	+
*Salmonella* sal19	25	26	17	22	32	20	21	27	8	22	23	19	+	+	+	+	+
*Salmonella* sal20	22	18	16	21	19	20	18	19	10	18	26	20	+	+	+	+	+
*Salmonella* sal21	26	24	18	21	25	21	24	22	9	23	24	18	+	+	+	+	+
*Salmonella* sal21	22	24	19	21	30	21	22	19	9	21	22	19	+	-	+	+	-
*Salmonella* sal23	25	24	18	22	32	22	22	18	9	22	28	18	+	+	+	+	+
*Salmonella* sal24	29	26	21	22	31	21	23	20	8	22	24	19	+	-	+	+	+
*Salmonella* sal25	26	24	18	21	30	21	23	20	9	22	23	17	+	+	+	+	+
*Salmonella* sal26	21	19	18	21	15	21	31	21	15	18	8	21	+	-	+	+	+
*Salmonella* sal27	22	26	17	18	17	12	21	27	8	22	23	19	+	-	+	+	+
*Salmonella* sal28	12	14	13	14	9	12	20	9	8	10	14	13	+	+	+	+	+
*Salmonella* sal29	26	24	18	23	30	22	26	22	9	22	24	19	+	-	+	-	+
*Salmonella* sal30	26	25	16	21	29	20	24	19	9	22	26	18	+	+	+	+	+
*Salmonella* sal31	19	20	18	17	14	19	17	22	9	19	16	18	+	+	+	+	+
*Salmonella* sal32	18	16	17	18	18	16	20	19	9	19	22	19	+	-	+	+	+
*Salmonella* sal33	24	24	18	22	32	22	22	18	9	22	28	18	+	-	+	+	+
*Salmonella* sal34	26	22	21	22	31	21	23	20	8	22	24	19	+	+	+	+	+
*Salmonella* sal35	26	24	18	21	30	21	23	20	9	22	23	17	+	+	+	-	+
*Salmonella* sal36	21	25	18	21	31	21	31	21	25	18	8	22	+	+	+	+	+
*Salmonella* sal37	25	18	17	22	32	14	18	16	8	22	23	12	+	+	+	-	+

## Data Availability

The only new data generated in this study relates to the genomic information of the bacteriophage, which is publicly accessible at NCBI (https://www.ncbi.nlm.nih.gov/nuccore/2814448250) (accessed on 9 October 2024).

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
