# Peer review of "Isolation and Characterization of a Bacteriophage with Potential for the Control of Multidrug-Resistant *Salmonella* Strains Encoding Virulence Factors Associated with the Promotion of Precancerous Lesions"

_viruses, 2024, doi:10.3390/v16111711_

Round 1

Reviewer 1 Report

Comments and Suggestions for Authors

Manuscript Review: “Isolation and characterization of a bacteriophage with potential for the control of multidrug-resistant Salmonella strains encoding virulence factors associated with the production of precancerous lesion”

The authors have described an important investigation into the utility of a bacteriophage to counter growth of Salmonella strains that are not only resistant to various antimicrobial compounds but also exhibit a propensity to cause precancerous lesions. As the authors accurately state, bacteriophage-based therapy is the leading alternative to antibiotics that are increasingly becoming ineffectual against a multitude of bacterial infections. Their efforts to delineate the activity of a single power bacteriophage against Salmonella is an important effort, and should be published. There are a handful of major and minor concerns (listed below) that should be addressed prior to publication.

MAJOR ISSUES:

1)    My biggest issue with the scientific study described in the manuscript is that the group chose to focus on a single bacteriophage as a means to control various Salmonella strains. Frequently, the therapeutic application of bacteriophages is conducted with multiple distinct phages in a “cocktail” to prevent the development of bacterial resistance to the phage treatment. The phages and bacteria have been locked in a 3-billion-year-old arms race against each other, and it is my fear that this single phage would be rapidly overcome with resistance by the bacteria with progressive use.

I realize that the study has been completed and that it is probably too late for changes to the investigation that was conducted. Thus, I think it would be useful for the authors of the manuscript to address this shortcoming in the manuscript in some fashion. For example, they should address why they selected for just a single phage from the wastewater samples. There must have been several phages that were detected during the screening process. They also address the fears of resistance against a single bacteriophage entity in their discussion/conclusions section.

2)    Page 4: one-step replication curve

a.     The authors need to explain the reasoning behind using chloroform for part of the experiment and/or cite a reference to the reasoning for this approach.

b.     The burst size calculation is missing from their description and how they assessed the “eclipse phase”, “adsorption phase”, and “latency phase”

MINOR ISSUES:

1)    The authors need to establish abbreviations, particularly those with respect to time. Ex: minutes = min, hours = h, plaque forming units = PFU, colony forming units = CFU, etc. And once the abbreviation is established, they should use that abbreviation thereafter. There are paragraphs where the authors go back and forth using minutes and then min

2)    On page 2, first paragraph:

a.     “tumorigenesis” has an accent mark that needs to be removed

b.     Salmonella’s role in…

3)    On page 2, second paragraph: …due to their highly species-specific manner, which enables…

4)    Page 5, second paragraph: the authors need to description what they mean by the “phage extension” protocol and/or provide a reference for the method

5)    There are an occasionally few instances where I believe Spanish is being used in the manuscript that needs to be changed to English. Example: page 4 “de” in the “one-step replication curve” paragraph. Example: page 11 “foro” in the “host range” paragraph.

6)    Starting at Page 10, the page numbering is off as this page is listed as Page 2

7)    The authors need to establish the “bacteriophage” = “phage” toward the beginning of the manuscript and use “phage” thereafter

Author Response

Reviewer's comment: The authors have described an important investigation into the utility of a bacteriophage to counter growth of Salmonella strains that are not only resistant to various antimicrobial compounds but also exhibit a propensity to cause precancerous lesions. As the authors accurately state, bacteriophage-based therapy is the leading alternative to antibiotics that are increasingly becoming ineffectual against a multitude of bacterial infections. Their efforts to delineate the activity of a single power bacteriophage against Salmonella is an important effort, and should be published. There are a handful of major and minor concerns (listed below) that should be addressed prior to publication.

Author response: Thank you for your thoughtful and encouraging comments regarding our investigation into the utility of the bacteriophage Phylax-28 against multi-drug-resistant Salmonella strains. We sincerely appreciate your recognition of the significance of our research, especially in the context of combating bacterial infections that pose substantial health risks, including the potential development of precancerous lesions.

We are fully committed to addressing all your major and minor concerns comprehensively to ensure that our manuscript meets the high standards required for publication.

We have worked diligently to incorporate your suggestions into our revised manuscript.

Once again, thank you for your valuable insights.

Reviewer's comment:  MAJOR ISSUES:

1)    My biggest issue with the scientific study described in the manuscript is that the group chose to focus on a single bacteriophage as a means to control various Salmonella strains. Frequently, the therapeutic application of bacteriophages is conducted with multiple distinct phages in a “cocktail” to prevent the development of bacterial resistance to the phage treatment. The phages and bacteria have been locked in a 3-billion-year-old arms race against each other, and it is my fear that this single phage would be rapidly overcome with resistance by the bacteria with progressive use.

I realize that the study has been completed and that it is probably too late for changes to the investigation that was conducted. Thus, I think it would be useful for the authors of the manuscript to address this shortcoming in the manuscript in some fashion. For example, they should address why they selected for just a single phage from the wastewater samples. There must have been several phages that were detected during the screening process. They also address the fears of resistance against a single bacteriophage entity in their discussion/conclusions section.

Author response: We sincerely thank the reviewer for raising important concerns regarding the selection of a single bacteriophage, Phylax-28, for this study. We greatly value the opportunity to address these points and to clarify our rationale.

First, we fully recognize the well-established concept of utilizing phage cocktails in therapeutic applications to prevent the emergence of phage-resistant bacteria. Our research group is actively continuing efforts to isolate additional bacteriophages from various environmental sources, and we foresee integrating these into a future formulation alongside Phylax-28. We agree that using multiple phages can enhance the long-term efficacy of bacteriophage therapies by reducing the risk of bacterial resistance. However, we believe that this study, focusing on the isolation and characterization of a single bacteriophage, still provides significant insights into phage therapy against multidrug-resistant Salmonella strains.

It is important to emphasize that while several bacteriophages were indeed detected during the screening process from wastewater samples, Phylax-28 was chosen for detailed characterization based on specific and notable attributes. This bacteriophage exhibited exceptionally large plaque sizes, as shown in Figure 1A, indicating robust lytic activity.  In addition, Phylax-28 in preliminary assays showed remarkable stability during storage, a key factor in evaluating its potential for therapeutic use. This combination of factors made Phylax-28 the most promising candidate for further study, and this decision was made after careful consideration of all available data. Thanks to the reviewer’s comment, we have clarified this reasoning more explicitly in the revised version of the manuscript.

Regarding the issue of potential bacterial resistance to a single bacteriophage, we acknowledge the concern and have included an expanded discussion in the manuscript’s conclusion. We outline the inherent risks of using a single phage entity and highlight ongoing research to develop phage cocktails. We also mention our future direction of isolating and characterizing additional phages that may complement Phylax-28 in combating Salmonella and reducing the risk of resistance.

We believe that our study remains aligned with the goals of Viruses, contributing valuable data on the therapeutic potential of bacteriophages for controlling multidrug-resistant pathogens.

Once again, we appreciate your valuable feedback, which has helped to refine and clarify the future directions of our research.

Reviewer's comment: MAJOR ISSUES:

2)    Page 4: one-step replication curve

  1. The authors need to explain the reasoning behind using chloroform for part of the experiment and/or cite a reference to the reasoning for this approach.
  2. The burst size calculation is missing from their description and how they assessed the “eclipse phase”, “adsorption phase”, and “latency phase”

Author response: Thank you very much for your insightful comments, which have greatly contributed to improving the clarity of our manuscript. We are pleased to address the points you raised.

  1. We appreciate your inquiry regarding the use of chloroform in our experiment. The use of chloroform is particularly significant, as it allows for the determination of phage replication at the intracellular level. By lysing the bacterial cells, chloroform releases phage particles produced within the host, enabling a more precise understanding of the phage lifecycle dynamics. We have added this explanation in the revised version of the manuscript.
  2. In response to your second comment, we have clarified in Section 2.5 of the revised manuscript how the burst size was calculated, as well as how the "eclipse phase," "adsorption phase," and "latency phase" were assessed during the experiment. This additional information should provide a more comprehensive description of the methodology. We greatly appreciate your constructive feedback and the opportunity to improve our manuscript.

Reviewer's comment: MINOR ISSUES

1)    The authors need to establish abbreviations, particularly those with respect to time. Ex: minutes = min, hours = h, plaque forming units = PFU, colony forming units = CFU, etc. And once the abbreviation is established, they should use that abbreviation thereafter. There are paragraphs where the authors go back and forth using minutes and then min

Author response: Thank you for your insightful comments regarding the use of abbreviations in our manuscript. We appreciate your attention to detail, which is essential for maintaining clarity and consistency in scientific writing.

In the revised version of the manuscript, we have ensured that all abbreviations, particularly those related to time and units, are clearly defined upon their first use. For example, we have established “minutes” as “min,” “hours” as “h,” “plaque-forming units” as “PFU,” and “colony-forming units” as “CFU.” After their initial introduction, we have consistently used these abbreviations throughout the text to avoid any confusion.

Thank you once again for your valuable feedback, which has helped enhance the quality of our manuscript.

Reviewer's comment: MINOR ISSUES

2)    On page 2, first paragraph:

  1. “tumorigenesis” has an accent mark that needs to be removed
  2. Salmonella’srole in…

Author response: Thank you for your careful review and constructive feedback regarding our manuscript. You are correct that the word “tumorigenesis” should not have an accent mark, and we appreciate you bringing this to our attention. Additionally, we have ensured that the phrasing “Salmonella’s role in…” has been appropriately addressed in the revised version of the manuscript.

We have made these corrections and appreciate your guidance in helping us enhance the quality of our work. Thank you once again for your valuable insights.

Reviewer's comment: MINOR ISSUES

3)    On page 2, second paragraph: …due to their highly species-specific manner, which enables…

Author response: Thank you for your insightful comments regarding our manuscript. You are correct in noting the phrasing "due to their highly species-specific manner, which enables…". We have addressed this observation in the revised version of the manuscript, ensuring that the text flows more smoothly and enhances the overall clarity. We appreciate your feedback, as it undoubtedly contributes to improving the quality of our work. Thank you once again for your valuable insights.

Reviewer's comment: MINOR ISSUES

4)    Page 5, second paragraph: the authors need to description what they mean by the “phage extension” protocol and/or provide a reference for the method

Author response: Thank you for your thoughtful comment regarding our use of the term "phage extension." We sincerely apologize for any confusion; we were unable to locate the expression "phage extension" and believe it may have been a misinterpretation. We intended to refer to the "plaque extension" method. However, to ensure clarity, we have corrected the term to "plate count method" in the revised manuscript. We appreciate your understanding and your valuable feedback, which has undoubtedly improved the clarity of our work. Thank you once again for your insights.

Reviewer's comment: MINOR ISSUES

5)    There are an occasionally few instances where I believe Spanish is being used in the manuscript that needs to be changed to English. Example: page 4 “de” in the “one-step replication curve” paragraph. Example: page 11 “foro” in the “host range” paragraph.

Author response: Thank you for your careful review of our manuscript and for pointing out the instances of Spanish text that required correction. You are correct; we have addressed these issues in the revised manuscript, ensuring that all text is appropriately presented in English. We appreciate your diligence in helping us enhance the clarity and professionalism of our work. Thank you once again for your valuable feedback.

Reviewer's comment: MINOR ISSUES

6)    Starting at Page 10, the page numbering is off as this page is listed as Page 2

Author response: Thank you for your meticulous attention to detail regarding the page numbering in our manuscript. You are correct that there was an error, and we have corrected the pagination in the revised manuscript to ensure it is accurate throughout. We appreciate your valuable feedback.

Reviewer's comment: MINOR ISSUES

7)    The authors need to establish the “bacteriophage” = “phage” toward the beginning of the manuscript and use “phage” thereafter

Author response: Thank you for your thoughtful feedback regarding the terminology used in our manuscript. In response to your comment, we have established the term "bacteriophages," also called phages, in the opening sections of the manuscript. We have revised the text to use "phage" in most instances; however, we retained the term "bacteriophage" where we deemed it necessary for clarity or specificity. This approach ensures that the terminology aligns with the context and enhances the overall readability of our work. We greatly appreciate your insights, which have undoubtedly contributed to improving the manuscript. Thank you once again for your valuable suggestions.

Reviewer 2 Report

Comments and Suggestions for Authors

In this manuscript Luis Amarillas presented isolation and characterization of a novel bacteriophage infecting Salmonella. The new phage Phylax-28 seems to be promising for phage therapy of infections caused by multidrug-resistant Salmonella strains. The study is technically sound and well presented. I have only few minor recommendations that could improve the quality of presentation.

 Please, include line numbers in the manuscript and correct page numbering

 page 4. “the bacteriophage was added at a multiplicity of infection (MOI) of 0.1”. How this value (cell titer) was calculated?

 page 5. “The Salmonella strain exhibiting the highest antibiotic resistance.” Strain ID should be given here.

 Table 3.  Abbreviations should be explained in a footnote

 3.3. Isolation of the bacteriophage

“with findings by 33 Jurczak-Kurek and coworkers,”  - provide a reference

3.4. Bacteriophage morphology

“non-contractile and rigid tail 8.2 ± 0.1 nm in length” I do not see any tail in Figure 1B. Please show it by an arrow.

 Figure 3B. In what units is the phage concentration is shown?  Pfu/ml?

Figure 6. Which “antibiotic,” and “probiotic bacteria”? Specify in the legend.

 3.9. Bacteriophage genomic analysis

I recommend adding a figure showing phage genome map.

Author Response

Reviewer's comment: In this manuscript Luis Amarillas presented isolation and characterization of a novel bacteriophage infecting Salmonella. The new phage Phylax-28 seems to be promising for phage therapy of infections caused by multidrug-resistant Salmonella strains. The study is technically sound and well presented. I have only few minor recommendations that could improve the quality of presentation.

Author response: Thank you very much for your positive feedback on our manuscript.  Your recognition of the technical soundness and presentation of our work is greatly appreciated. We also thank you for your recommendations, which we believe will further enhance the quality of our manuscript. We have carefully addressed each suggestion, and we trust that these revisions will meet your expectations. Once again, we sincerely appreciate your valuable insights and the time you have taken to review our work.

Reviewer's comment: Please, include line numbers in the manuscript and correct page numbering

Author response: Thank you for your attention to detail. We have added line numbers to the revised manuscript and corrected the page numbering accordingly. We hope this improves the clarity of the document.

Reviewer's comment: page 4. “the bacteriophage was added at a multiplicity of infection (MOI) of 0.1”. How this value (cell titer) was calculated?

Author response: We appreciate this valuable observation. In response, we have clarified the calculation of the MOI in the revised manuscript. In section 2.5 of the materials and methods, we now provide a detailed explanation of how both the bacteriophage titer and the bacterial cell concentration were determined to derive the MOI value of 0.1.

Reviewer's comment: page 5. “The Salmonella strain exhibiting the highest antibiotic resistance.” Strain ID should be given here.

Author response: Thank you for your valuable suggestion. In response to your comment regarding the strain identification, we have revised the manuscript accordingly. The sentence now reads: "The Salmonella strain, designated as Sal-28, which exhibited the highest level of antibiotic resistance and encoded all five evaluated virulence genes, was cultured in TSB medium at 37°C for 24 hours". We trust this clarification addresses your concern.

Reviewer's comment: Table 3.  Abbreviations should be explained in a footnote

Author response: Thank you for your observation. We have added a footnote to Table 3 explaining all abbreviations used in the table. The revisions are now included in the manuscript as follows: “IMP: Imipenem; SXT: Trimethoprim-sulfamethoxazole; GEN: Gentamicin; AMP: Ampicillin; CIP: Ciprofloxacin; NAL: Nalidixic acid; CHL: Chloramphenicol; TET: Tetracycline; COL: Colistin; AMC: Amoxicillin-clavulanate; CFP: Cefoperazone; AMK: Amikacin.”

 Reviewer's comment: 3.3. Isolation of the bacteriophage

“with findings by 33 Jurczak-Kurek and coworkers,”  - provide a reference

Author response: We appreciate your careful review and valuable suggestions. In response, we have added the appropriate reference to Jurczak-Kurek and coworkers in the revised version of the manuscript to support the argumentation in section 3.3.

Reviewer's comment: 3.4. Bacteriophage morphology

“non-contractile and rigid tail 8.2 ± 0.1 nm in length” I do not see any tail in Figure 1B. Please show it by an arrow.

Author response: Thank you for your valuable observation. In response to your comment, we have included a clear indication of the bacteriophage's tail in Figure 1B by marking it with a red arrow to enhance visibility and address the concern regarding its identification. We trust this modification clarifies the structural feature.

Reviewer's comment: Figure 3B. In what units is the phage concentration is shown?  Pfu/ml?

Author response: Thank you for your insightful observation. In response to your comment, we would like to clarify that the phage concentration in Figure 3B is expressed as the base-10 logarithm of PFU/ml. This has been indicated in the revised figure legend for clarity.

Reviewer's comment: Figure 6. Which “antibiotic,” and “probiotic bacteria”? Specify in the legend.

Author response: Thank you for your valuable comment. In response, we have specified that the antibiotic used in Figure 6 is ciprofloxacin. Additionally, the probiotic bacteria mixture includes five strains: Bacillus clausii, Bacillus coagulans, Lactobacillus paracasei, Lactobacillus rhamnosus, and Limosilactobacillus reuteri, mixed in equal proportions. This information has been included in the revised figure legend for clarity.

Reviewer's comment: 3.9. Bacteriophage genomic analysis

I recommend adding a figure showing phage genome map.

Author response: Thank you for this valuable recommendation. We have included a genomic map of the bacteriophage Phylax-28 in the revised manuscript, which provides a detailed schematic representation of the open reading frames (ORFs) and their corresponding functions.

Reviewer 3 Report

Comments and Suggestions for Authors

The manuscrip describes the isolation and characterization of a phage infecting multidrug-resistant Salmonella strains. While most of the characteristics of the phage are properly analyzed, I identified two parts which should be analyzed.

 • The authors wrote: " The virions exhibit an icosahedral structure, characterized by a capsid 55 composed of 20 equidistant triangular faces." This is a bit misleading. A short tail is clearly visible on a virion in figure 1B. the authors should mention that is a T7-like phage, with a podovirus morphology. They should identify the tail with an arrow.

• The authors wrote: "The bacteriophage genome was sequenced and analyzed using bioinformatics tools." Although they mention the number of genes identified in the various functions, they should provide a figure with the schematically representing the ORFs along the genome, and their function, or a table with the list of ORFs  and their function.

Author Response

Reviewer's comment: The manuscrip describes the isolation and characterization of a phage infecting multidrug-resistant Salmonella strains. While most of the characteristics of the phage are properly analyzed, I identified two parts which should be analyzed.

Author response: Thank you for your thoughtful feedback regarding our manuscript on the isolation and characterization of a phage infecting multidrug-resistant Salmonella strains. We appreciate your insights and will ensure that the two identified parts are thoroughly analyzed in the revised version. Your suggestions will help enhance the overall quality of the manuscript.

Reviewer's comment: The authors wrote: " The virions exhibit an icosahedral structure, characterized by a capsid 55 composed of 20 equidistant triangular faces." This is a bit misleading. A short tail is clearly visible on a virion in figure 1B. the authors should mention that is a T7-like phage, with a podovirus morphology. They should identify the tail with an arrow.

Author response: Thank you for your insightful comment. While the Autographiviridae and Podoviridae families may appear morphologically similar and potentially indistinguishable at first glance, genomic analysis confirms that the Phylax-28 bacteriophage belongs to the Autographiviridae family. Notably, members of the Autographiviridae encode their own RNA polymerase, as is the case with Phylax-28. In response to your suggestion, we have also illustrated this in Figure 7, which presents the genomic map of Phylax-28. Additionally, we have included red arrows in Figure 1B to clearly indicate the presence of the short tail on the virion. Your feedback has been invaluable in improving the manuscript.

Reviewer's comment: The authors wrote: "The bacteriophage genome was sequenced and analyzed using bioinformatics tools." Although they mention the number of genes identified in the various functions, they should provide a figure with the schematically representing the ORFs along the genome, and their function, or a table with the list of ORFs  and their function.

Author response: Thank you for your valuable observation. In response to your comment, we have included a figure in the revised manuscript that graphically represents the genome of the Phylax-28 bacteriophage. This figure details the locations of the open reading frames (ORFs) along the genome and describes their respective functions. Your suggestion has greatly enhanced the clarity and comprehensiveness of our genomic analysis.

Round 2

Reviewer 3 Report

Comments and Suggestions for Authors

The authors corrected the paper according to my suggestions